# Peripheral Neuropathy in Patients Recovering from Severe COVID-19: A Case Series

**DOI:** 10.3390/medicina58040523

**Published:** 2022-04-08

**Authors:** Pietro Balbi, Annamaria Saltalamacchia, Francesco Lullo, Salvatore Fuschillo, Pasquale Ambrosino, Pasquale Moretta, Bernardo Lanzillo, Mauro Maniscalco

**Affiliations:** 1Neurological Rehabilitation Unit of Telese Terme Institute, Istituti Clinici Scientifici Maugeri IRCCS, 82037 Telese Terme, Italy; pbalbi@dongnocchi.it (P.B.); annamaria.saltalamacchia@icsmaugeri.it (A.S.); francesco.rullo@icsmaugeri.it (F.L.); pasquale.moretta@icsmaugeri.it (P.M.); bernardo.lanzillo@icsmaugeri.it (B.L.); 2Fondazione Don Carlo Gnocchi ONLUS, Polo Riabilitativo del Levante Ligure, 19125 La Spezia, Italy; 3Respiratory Rehabilitation Unit of Telese Terme Institute, Istituti Clinici Scientifici Maugeri IRCCS, 82037 Telese Terme, Italy; salvatore.fuschillo@icsmaugeri.it; 4Cardiac Rehabilitation Unit of Telese Terme Institute, Istituti Clinici Scientifici Maugeri IRCCS, 82037 Telese Terme, Italy; pasquale.ambrosino@icsmaugeri.it; 5Department of Clinical Medicine and Surgery, University Federico II, 80138 Naples, Italy

**Keywords:** COVID-19, SARS-CoV-2, neuropathy, rehabilitation, disability, exercise, occupational medicine, chronic disease, outcome

## Abstract

*Background and Objectives*: Neurological manifestations have been reported in a significant proportion of coronavirus disease 2019 (COVID-19) patients. We aimed to evaluate the prevalence and severity of peripheral nervous system (PNS) involvement in a large group of convalescent COVID-19 patients undergoing in-hospital multidisciplinary rehabilitation. *Materials and Methods*: Convalescent COVID-19 patients admitted to a Pulmonary Rehabilitation Unit were consecutively screened for inclusion within 48 h of discharge from an acute care setting. All included patients underwent electrophysiological examinations. *Results*: Among 102 enrolled patients (mean age 62.0 years, 82.4% males), PNS electrophysiological alterations were detected in 42.2%. Mononeuropathies exclusively involving the peroneal nerve were observed in 8.8% (*n* = 9), while multiple mononeuropathies were similarly reported in nine patients (8.8%). A symmetric sensorimotor polyneuropathy was documented in 24.5% of participants (*n* = 25). A significant difference was found for exercise capacity and pulmonary function in post hoc comparisons between the three study groups. *Conclusions*: The risk of neuropathy in the convalescent phase of COVID-19 is relevant. This should be considered when planning multidisciplinary rehabilitation strategies.

## 1. Introduction

Coronavirus disease 2019 (COVID-19) is caused by severe acute respiratory syndrome coronavirus 2 (SARS-CoV-2) [1]. SARS-CoV-2 was initially thought to cause a respiratory disease, with severity ranging from absent or flu-like symptoms to a severe form of interstitial pneumonia often requiring intensive care admission due to the need for mechanical ventilation and high flow oxygen therapy. However, besides the respiratory system, other organs can be directly or indirectly targeted by SARS-CoV-2, giving rise to a composite and pleiotropic puzzle of symptoms [2]. In addition, a relevant number of COVID-19 patients may manifest the so-called “post-COVID-19 syndrome”, consisting of different and debilitating multi-organ symptoms (shortness of breath, fatigue, joint pain, chest pain, “brain fog”, intolerance orthostatic) that may persist for weeks or months after fighting off the infection [3,4]. In a significant proportion of COVID-19 patients a broad and increasing spectrum of neurological manifestations including encephalopathies, central nervous system vasculitis, cerebrovascular events and peripheral nervous system (PNS) involvement have been reported [5,6,7,8].

Data from Wuhan, China, indicate that 36.4% of COVID-19 patients present neurological manifestations, while in patients with severe disease the neurologic manifestations are observed in up to 45.5% [9]. Although the neurological manifestations of COVID-19 can result in significant disability, there is currently scant information available on their incidence, severity and patient risk factors. One of the most debilitating symptoms in post-COVID-19 syndrome is fatigue, which can be explained by muscle abnormalities, consistent with the high creatine phosphokinase (CPK) titles frequently present after the acute phase [10,11]. However, fatigue in addition to the poorly localized pain and numbness that are often reported by COVID-19 patients, suggest the involvement of the PNS. Indeed, growing evidence links COVID-19 to PNS manifestations. In a study by Malik et al., conducted on 83 COVID-19 patients hospitalized with acute respiratory distress syndrome (ARDS), PNS involvement was reported in 12 (14.5%), and was always related to the prone position of these patients [12]. Similar results were observed by Frithiof et al. [13].

The manifestations of PNS involvement in severe COVID-19 are likely the result of several mechanisms, including critical illness neuropathy, systemic inflammatory state, metabolic alterations, neurotoxic side effects of drugs used to treat COVID-19, and to a lesser extent, from the compression of nerves after prolonged hospitalization in intensive care [14]. In the general population, the outcome of PNS injury is often unsatisfactory and, in survivors of severe COVID-19, due to the frequent presence of certain risk factors such as diabetes, obesity and older age, the risk of poor outcome is probably greater [12,13,15]. Therefore, an early diagnosis and symptomatic treatment of PNS involvement in the early stages of the infection would be useful to allow the patient to overcome the acute phase of the infection and to prevent the onset of SARS-CoV-2 post-acute burden.

In the present study, we aim to evaluate the prevalence and severity of PNS involvement in a large group of convalescent COVID-19 patients undergoing in-hospital multidisciplinary rehabilitation.

## 2. Materials and Methods

From January 2021 to September 2021, convalescent COVID-19 patients admitted to the Pulmonary Rehabilitation Unit of Istituti Clinici Scientifici Maugeri Spa SB, IRCCS of Telese Terme, were consecutively screened for inclusion within 2 months from swab test negativization. The inclusion criteria were: recent SARS-CoV-2 infection confirmed by reverse transcription polymerase chain reaction (RT-PCR) of the nasopharyngeal swab; history of severe COVID-19 according to World Health Organization (WHO) classification; two negative nasopharyngeal swab tests for SARS-CoV-2 (spaced 1 week apart) before discharge from the sub-intensive care unit. The exclusion criteria were: diabetes; presence of premorbid neuropathy or premorbid symptoms compatible with diagnosis of neuropathy; alcohol abuse; cancer; family history of genetic neuropathy; CPK values beyond the normal range (30–200 U/L). After approval of the study protocol by the local Institutional Review Board (reference number ICS 11/20), all included patients provided a written informed consent to use their de-identified data. Whenever applicable, the Strengthening the Reporting of Observational Studies in Epidemiology (STROBE) recommendations were followed [16].

### 2.1. Main Study Procedures

After informed consent signature, the main demographic and clinical information pertaining to the acute phase of COVID-19, pulmonary function, and exercise capacity were collected for all convalescent COVID-19 patients. Patients performed pulmonary function tests (PFTs), including spirometry, carbon monoxide diffusion capacity (DLCO), and measurement of static lung volumes. PFTs were performed using automated equipment (Vmax^®^ Encore, Vyasis Healthcare, Milan, Italy) according to American Thoracic Society/European Respiratory Society (ATS/ERS) guidelines [17,18]. Arterial blood gas analysis, if the patient’s clinical conditions allowed, was performed in a sitting position by breathing ambient air for at least 30 min, using a blood gas analyzer (ABL 825^®^ FLEX BGA, Radiometer Medical Aps, Copenhagen, Denmark). In case of severe resting hypoxemia, blood gas analysis was performed while the patient was on oxygen therapy. Exercise capacity was evaluated with the 6-min walking test (6MWT) in accordance with the ATS guidelines [19]. The 6-min walking distance (6MWD) covered at the end of 6MWT was measured in meters, whereas the perceived dyspnea and fatigue were rated by a modified Borg category ratio 10 scale [20].

### 2.2. Electrophysiological Examination

Peripheral nerve conduction examination was performed using a Dantec Keypoint system, by Natus Neurology (Alpine Biomedical Aps, Denmark). All patients had a standard battery of nerve conduction motor and sensory studies, including median, ulnar, radial, peroneal, sural and tibial nerves on both sides, using both recording and stimulating surface electrodes (Ambu, Denmark), with a bandwidth of 5–10 kHz for motor nerve conduction, and 10 Hz to 2 kHz for sensory nerve conduction. Following electrical stimuli of the different nerves, motor nerve conduction was examined by recording the compound muscle action potential (CMAP) from surface electrodes overlying the muscle supplied by each motor nerve. Following a supramaximal stimulus, reproducible values of peak-to-peak CMAP amplitude (mV) and latency (ms) from the stimulus to the onset of the CMAP were recorded. Each nerve was stimulated at distal and proximal sites, and the fastest motor nerve conduction velocity was calculated (m/s).

Motor conduction data were collected in the following nerves of both sides: median (stimulation at the wrist and elbow, recording from the abductor pollicis brevis muscle), ulnar (stimulation at the wrist, the forearm and the elbow, recording from the abductor digiti minimi muscle), peroneal (stimulation at the ankle and the fibula head, recording from the extensor digitorum brevis muscle), and tibial nerve (stimulation at the ankle and the knee, recording from the abductor hallucis muscle). Sensory nerve action potentials (SNAPs) were obtained by antidromically testing multiple nerves. The sensory latency and the onset-to-peak amplitude of the SNAPs were measured, and an antidromic sensory nerve conduction velocity was calculated for each nerve.

Sensory conduction data were collected in the following nerves of both sides: median (stimulus at the wrist, recording from the third digit), ulnar (stimulus at the wrist, recording from the fifth digit), radial (stimulation at the wrist, recording from first digit), and sural nerve (stimulation at the lateral middle lower leg, recording from the ankle). According to the electrophysiological results, four categories of peripheral nerve involvement could be identified: Category 1—no peripheral abnormalities (normal); Category 2—mononeuropathy, when abnormalities were found only on a single nerve; Category 3—multiple mononeuropathy, when scattered, non-symmetrical abnormalities were found in multiple nerves; Category 4—symmetrical polyneuropathy.

### 2.3. Statistical Analysis

Continuous data were expressed as mean ± standard deviation (SD). Categorical variables were summarized as relative frequencies and analyzed by means of a Chi-square test. Data from the different groups were analyzed using non-parametric Kruskal–Wallis test. Post hoc analyses were performed using Mann–Whitney analyses. A two-tailed alpha level of 0.05 was adopted in all analysis with the exception of multiple comparisons, for which the Bonferroni’s correction was applied. The commercially available software IBM SPSS Statistics 24.0 system (SPSS Inc., Chicago, IL, USA) was used for all statistical analyses. 

## 3. Results

Among 125 patients screened for eligibility, 11 refused to enter the study for personal reasons, five were ineligible for protocol adherence issues and seven for poor compliance in the study procedures. Therefore, a total of 102 patients were included in the present study. Their average age was 62.03 ± 10.04 and 82.4% were men. Demographic and clinical features of the study population are shown in Table 1.

Among the 102 enrolled patients, PNS electrophysiological alterations were detected in 42.2%, while 57.8% (*n* = 59) were normal (Category 1). The results of the electrophysiological tests showed abnormalities in the amplitude of severity from mild to moderate, while the conduction velocities were only slightly compromised. The mononeuropathies consisted of low amplitude CMAPs exclusively involving the peroneal nerve (Category 2), and were observed in 8.8% (*n* = 9). Multiple mononeuropathies with asymmetric abnormalities involving multiple nerves (Category 3) were also reported in 8.8% of patients (*n* = 9), with the main electrophysiological abnormalities detected in these cases being the amplitude reductions in both motor and sensory potentials. Finally, symmetric sensorimotor polyneuropathy (Category 4) was observed in 24.5% of patients (*n* = 25). Given the relatively small number of subjects in the categories 2 and 3, we grouped them for statistical purposes and considered the new category as intermediate in severity between normal and polyneuropathy classes (Table 1). As shown in Figure 1, a significant difference was found for exercise capacity and pulmonary function in post hoc comparisons between the three study groups (normal vs. mononeuropathy vs. polyneuropathy). In particular, patients without PNS involvement exhibited a significantly higher 6MWD when compared to both mononeuropathy (205.52 ± 103.84 m vs. 117.76 ± 116.94 m) and polyneuropathy patients (205.52 ± 103.84 m vs. 147.85 ± 64.98 m). Similar findings were obtained when considering values of partial pressure of oxygen in arterial blood (PaO_2_) in the three study groups (77.86 ± 13.74 mmHg, 68.32 ± 14.98 mmHg, and 62.15 ± 10.61 mmHg, respectively). Patients without electrophysiological alterations also showed significantly higher values of FVC (79.53 ± 17.97 L vs. 65.94 ± 14.71 L) and DLCO (61.51 ± 18.73% of predicted vs. 39.0 ± 15.71% of predicted) as compared to those with polyneuropathy. No significant difference between the different categories of PNS involvement was found when considering parameters related to disease severity, including WHO class, mechanical ventilation and high-flow oxygen therapy during the acute phase.

## 4. Discussion

In this study, we report that a large proportion of patients affected by COVID-19 (42%), following severe manifestations of the acute illness worthy of in-hospital management, show an involvement of the PNS, when examined while recovering in a rehabilitation setting. Thus, the study is a preliminary report on the raw prevalence of peripheral nerve involvement in the most severe cases of SARS-CoV-2 infection.

A few recent studies reported on the presence of peripheral nerve involvement in SARS-CoV-2. Most of them considered only the acute phase of the illness, wherein critical illness neuromyopathy [21,22,23], Guillain–Barré syndrome [22,24,25], polyneuropathy [22], multiplex mononeuropathy [26] or muscle damage and nerve pain [6] were described. Fewer reports extended the presence of peripheral nerve involvement beyond the acute illness, during the manifestations of the post-COVID-19 syndrome [27]. These studies, usually consisting of small case series reports, provided discordant results about the type of peripheral nerve involvement, ranging from demyelinating polyneuropathy with myopathy [27] to multiple axonal mononeuropathy [15]. It is worth noting, in addition, that a lasting neurodegeneration and neuropathy in patients surviving the acute infection has also been described in other coronavirus diseases [28]. Although SARS-CoV-2 primarily affects the respiratory system, several case reports and case series have recently documented the clinical relevance of neurological manifestations in COVID-19 patients. In some cases, neurological manifestations represent the presenting symptom of SARS-CoV-2 infection, even in the absence of fever and respiratory involvement [22].

The involvement of central and peripheral nervous systems both contribute to the composition of the clinical picture of COVID-19. However, the available literature data on neurological manifestations in COVID-19 patients, particularly those involving the PNS, are still scarce and, for various reasons, discordant. Furthermore, from the current epidemiological knowledge it is not possible to establish with certainty whether and to what extent the neurological injury is caused by SARS-CoV-2 or is the result of a critical illness. Rufino et al., in a study conducted on 1760 patients with COVID-19, reported that 137 (7.8%) developed clinically evident PNS involvement: 17 Guillain–Barré Syndrome (GBS), nine Critical Illness Myopathy and Neuropathy (CRIMYNE), two brachial plexopathies, and three peripheral polyneuropathies (PNP) [22]. In a study conducted by Mao et al., on 214 patients with COVID-19, 36.4% subjects reported neurological symptoms, of which 8.9% had PNS manifestations [6]. The analysis of data from the ALBACOVID registry found that 57.4% of COVID-19 patients had neurological symptoms [29]. Although PNS involvement is known to contribute significantly to long-term disability in survivors of severe COVID-19, only a few studies have assessed its prevalence. Malik et al., first reported 12 COVID-19 survivors admitted to rehabilitation with PNS damage due to prone position [12]. Stuart-Neto et al., found that 6.7% of COVID-19 patients with severe respiratory conditions developed peripheral neuropathy [30]. PNS involvement related to COVID-19 generally shows a symmetrical pattern. However, it has been reported that in 16% of COVID-19 survivors in addition to symmetrical weakness, possibly related to sarcopenia, severe focal neurological deficits are present related to superimposed mononeuropathies [15].

In the present study, we report on the results of the neurophysiological tests performed on a large series of 102 patients who survived a severe form of COVID-19. The study, conducted in a rehabilitation setting, showed that in a large percentage (42.5%) of these patients, when studied during the convalescence phase, electrophysiological alterations can be found compatible with a peripheral neuropathy. Our results are in line with those reported by Franz et al., that diagnosed 63 peripheral nerve abnormalities in 32 patients who survived severe COVID-19 [31]. As compared to other studies that investigated the involvement of PNS in subjects recovered from COVID-19, we found a significantly higher percentage of affected patients. It is worth mentioning that those patients who had an isolated, even bilateral, electrophysiological abnormality of the median nerve on the wrist were highly suggestive of this syndrome and were not included in our mononeuropathy group, due to the large incidence of carpal tunnel syndrome in the general population. Unfortunately, due to the lack of a non-COVID-19 control group matched for clinical and demographic features, we cannot exclude that the neurophysiological alterations documented in our study population may be the result of a prolonged hospitalization rather than the consequence of a specific, direct or indirect noxious effect of the novel coronavirus on peripheral nerves [32,33].

The pathophysiology of the different neurological features caused by COVID-19 has yet to be fully understood. Peripheral nervous system disease secondary to COVID-19 likely occurs as a result of multiple pathological processes that include endovascular dysfunction, mechanical injury (i.e., compression of nerve trunks following prolonged bedding in the Intensive Care Unit), neurotoxic drugs, inflammatory disease, oxidative stress and apoptosis [23,28]. Vasculitis and the hypercoagulable state seen in COVID-19, leading to thrombosis within the vasa nervorum and resulting in peripheral neuropathy, have been described.

Proposed mechanisms relate to immune dysregulation, complement activation, clotting pathway activation or viral dissemination with direct systemic endothelial infection [23]. A report on histopathologic findings in the skeletal muscle and peripheral nerve from 35 consecutive autopsies performed on patients with COVID-19 did not find any signs of direct viral invasion of muscle and nerve fibers [34]. The study, instead, demonstrated muscle and nerve tissue inflammatory/immune-mediated damage likely related to release of cytokines. Another mechanism may be an aggression of PNS by an immune-mediated mechanism. This mechanism is known as “molecular mimicry”. SARS-CoV-2 has two hexapeptides in common with human shock protein: proteins 90 and 60. Both of them have immunogenic potentials, and they are among the 41 human proteins associated with Guillain–Barrè syndrome and chronic inflammatory demyelinating polyneuropathy [35]. Furthermore, in a study of Stephen Keddie, no association was found between GBS and COVID-19, but a reduction in GBS incidence due to lockdown measure reducing transmission of GBS caused by pathogens such as Campylobacter Jejuni and respiratory viruses [36].

The SARS-CoV-2 has a higher affinity for angiotensin-converting enzyme receptor 2 (ACE-2) that is expressed on endothelial cells and neurons. This explains a higher neuro-invasive capacity of SARS-CoV-2 as compared with previous coronaviruses [37].

Neurological manifestations appear to be more frequent in patients who have developed severe disease and in the elderly. In our study group, the damage of the PNS was not correlated with either the WHO disease severity score, the need for mechanical ventilation, and the need for high-flow oxygen therapy; the latter two being indirect indicators severity of the disease.

However, a higher age, hypoxemia severity, as well as lower DLCO values were found in patients manifesting severe PNS as compared to subjects without neurophysiological alterations. This is in line with the evidence of chronic hypoxia and reduced DLCO as risk factors for peripheral neuropathy, with electrophysiological abnormalities being reported in about 95% of hypoxic subjects with chronic obstructive pulmonary disease (COPD) [38].

It has been reported that the length of ICU stay seems to be the most relevant risk factor for the appearance of PNS complications. However, in our case series, although the length of hospital stay was longer in patients with PNS involvement than in subjects with normal electrophysiological tests, the difference was not statistically significant.

In the present study, we showed that exercise capacity was somehow related to the presence of PNS abnormalities, similarly to what has been observed in COPD patients [39], since participants without PNS involvement reported significantly higher values of 6MWD when compared to both the polyneuropathy and mononeuropathy groups. However, considering that the normal patient group was also characterized by a lower age and higher DLCO and PaO_2_ values, we cannot exclude that the better exercise capacity may depend on these three variables, rather than the presence of PNS involvement.

Furthermore, the present study has other limitations. This study is a single rehabilitation center setting-based, with a relatively small number of patients so it does not necessarily reflect the incidence of COVID-19 peripheral nerve complications in the community. In addition, this study is without a control group of patients discharged for other reasons. Without an appropriate control group it remains unclear whether there is a causative relationship between COVID-19 and the reported PNS abnormalities. Finally, we cannot exclude that we enrolled patients with pre-existing neuropathy. However, in order to reduce this risk, we excluded patients with factors potentially causing neuropathy, such as diabetes, ascertained neuropathy or presence of premorbid symptoms compatible with neuropathy. Muscular involvement was not investigated in this study. However, a recent review showed that the incidence of muscular involvement (myositis or rhabdomyolysis was very low (eight case reports and nine patients). Myalgia was very frequent, although it is often only a symptom that we do not consider since it could be due to the psychological status of patients, i.e., not objective [40].

## 5. Conclusions

In conclusion, our neurophysiological study conducted during the rehabilitation phase on COVID-19 patients showed a frequent and underestimated peripheral neuropathy. In future, these results should be compared with a follow up study and/or in a multicentric approach.

## Figures and Tables

**Figure 1 medicina-58-00523-f001:**
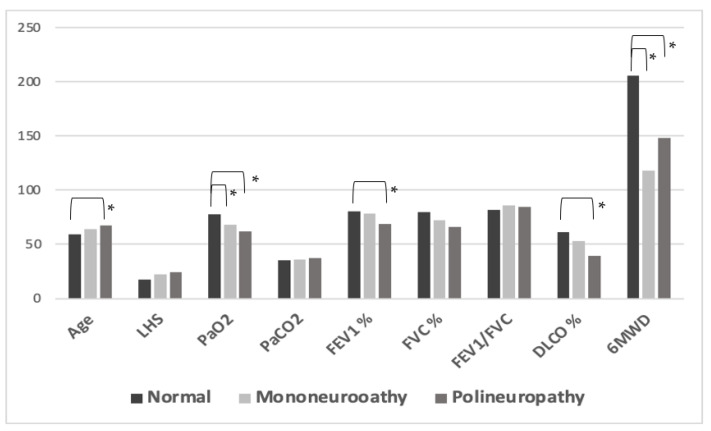
Post hoc comparisons between different categories of convalescent coronavirus disease 2019 (COVID-19) patients based on severity of neuropathy. LHS: length of hospital stays; WHO: World Health Organization; PaO_2_: partial pressure of oxygen in arterial blood; PaCO_2_: partial pressure of carbon dioxide; FEV_1_: forced expiratory volume in 1 s; FVC: forced vital capacity; 6MWT: six-minute walking test; DLCO: diffusion lung capacity for carbon monoxide. * Significant difference between groups at alpha level set at 0.05. Post-hoc comparisons were performed by means of Mann–Whitney analysis.

**Table 1 medicina-58-00523-t001:** Clinical characteristics of convalescent coronavirus disease 2019 (COVID-19) patients based on severity of neuropathy.

Clinical Variables	Total Sample *n* = 102	Normal *n* = 59	Mononeuropathy *n* = 18	Polyneuropathy *n* = 25
Age, years	62.5 (40–87)	59 (40–80)	62.5 (48–86)	67 (52–87)
Males, *n* (%)	84 (82.4)	47 (79.7)	15 (83.3)	22 (88)
Hypertension, *n* (%)	49 (48)	26 (44.1)	8 (44.4)	15 (60)
Hypercholesterolemia, *n* (%)	5 (5.9)	1 (1.7)	3 (16.7)	2 (8)
History of cardiovascular events, *n* (%)	12 (11.8)	4 (6.8)	3 (16.7)	5 (20)
Obesity, *n* (%)	24 (23.5)	3 (5.1)	9 (50)	12 (48)
History of neurological events, *n* (%)	9 (8.8)	1 (1.7)	4 (22.2)	4 (16)
LHS (days)	19 (0–68)	18 (0–50)	21 (0–48)	24 (0–68)
WHO, class of severity III/IV (*n*)	39/63	19/40	10/8	10/15
Non invasive ventilation, *n* (%)	30 (29.4)	19 (32.2)	4 (22.2)	7 (28)
High flow oxygen therapy, *n* (%)	26 (25.5)	14 (23.7)	7 (38.9)	5 (20)
PaO_2_, mmHg	71 (44–99)	79 (49–99)	66.7 (47–91)	61.3 (44–80)
PaCO_2_, mmHg	36 (25–46)	36 (25–43)	35.5 (29–43)	37 (25–46)
FEV_1_, %predicted	75 (34–114)	79 (34–114)	74.5 (54–104)	64 (53–102)
FVC, %predicted	76 (39–113)	78 (39–113)	69 (50–100)	68.5 (44–86)
FEV_1_/FVC	84 (40–100)	83.5 (40–100)	86 (79–91)	85 (72–93)
DLCO, %predicted	51.5 (20–110)	57 (28–110)	46 (25–99)	32 (20–75)
6MWD, metres	208 (0–365)	225 (0–365)	75 (0–338)	177 (30–210)

*n*: number; LHS: length of hospital stays; WHO: World Health Organization; PaO_2_: partial pressure of oxygen in arterial blood; PaCO_2_: partial pressure of carbon dioxide; FEV_1_: forced expiratory volume in 1 s; FVC: forced vital capacity; DLCO: diffusion lung capacity for carbon monoxide; 6MWD: six-minutes walking distance; Continuous data are reported as medians (minimum-maximum) unless otherwise indicated. Alpha level set at *p* = 0.005 according to Bonferroni’s correction. The comparisons were performed by means of Kruskal–Wallis test.

## Data Availability

Not applicable.

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
