# Peer review of "Peripheral Neuropathy in Patients Recovering from Severe COVID-19: A Case Series"

_medicina, 2022, doi:10.3390/medicina58040523_

Round 1

Reviewer 1 Report

I can propose to authors to include in their first table some more relevant parameters such as comorbidities, certain treatments in hospital as well common relevant post-Covid syndromes such as PE. Generally, the table 1 needs improvement e.g. omit the p-values because have no sense and present only the pairwise comparisons. I noticed that the summary statistics in numerical variables are presented in means & SD.... However, the authors used non-parametric comparison tests (see statistics paragraph). This is a conflict case... If the distributions are normal, then we use mean/SD and parametric comparison tests (ANOVA & t-test) or the data are not normally distributed and should use median (min-max) and non-parametric comparison tests. Please clarify and correct.

As we know the case-series studies have limited capacity to confirm research hypotheses bur are suitable to generate hypotheses. Then I propose to authors to focus their discusion in that direction by analysing their data in relation with the bibliography to the major point.... Are neuropathies consequences of the virus itself or it is the critical illness the main reason. That discussion will attract remarkable attention of the readers.

I am not happy with the strong conclusion (not supported by the data) that respiratory physiology predicts neuropathy (see Bradford Hill criteria for causation). Actually, we do not know if the observed differences among the groups are related to the disease or they are already pre-existed. I propose to authors to stay not so strong in their conclusions considering the small numbers and the above limitations. These limitations must be formulated in small paragraph in discussion section.

Author Response

I can propose to authors to include in their first table some more relevant parameters such as comorbidities, certain treatments in hospital as well common relevant post-Covid syndromes such as PE. Generally, the table 1 needs improvement e.g. omit the p-values because have no sense and present only the pairwise comparisons. I noticed that the summary statistics in numerical variables are presented in means & SD.... However, the authors used non-parametric comparison tests (see statistics paragraph). This is a conflict case... If the distributions are normal, then we use mean/SD and parametric comparison tests (ANOVA & t-test) or the data are not normally distributed and should use median (min-max) and non-parametric comparison tests. Please clarify and correct.

REPLY: Thank you for pointing out this relevant aspect. We modified the table 1 by changing statistics (median and min-max) as suggested by reviewer. According to reviewer 1 suggestion, we removed p-values from Table 1. Post-hoc comparisons have been graphically represented in Figure 1.

As we know the case-series studies have limited capacity to confirm research hypotheses bur are suitable to generate hypotheses. Then I propose to authors to focus their discussion in that direction by analysing their data in relation with the bibliography to the major point.... Are neuropathies consequences of the virus itself or it is the critical illness the main reason. That discussion will attract remarkable attention of the readers.

REPLY: We have modified part of discussion following the Reviewer’s suggestions. We hope that the discussion is now more attractive.

 I am not happy with the strong conclusion (not supported by the data) that respiratory physiology predicts neuropathy (see Bradford Hill criteria for causation). Actually, we do not know if the observed differences among the groups are related to the disease or they are already pre-existed. I propose to authors to stay not so strong in their conclusions considering the small numbers and the above limitations. These limitations must be formulated in small paragraph in discussion section.

REPLY: We agree with the Reviewer. We have deleted some sentences from conclusions and from the abstract and added some sentences to the limitations.

Reviewer 2 Report

I thank the editor for the possibility of reviewing the article entitled: "PERIPHERAL NEUROPATHY IN PATIENTS RECOVERING FROM SEVERE COVID-19: A CASE SERIES". The article is very interesting, however, the authors must review some points.

  1. In the methods section, add the reference of the STROBE checklist
  2. In table 1, the values of PaO2, DLCO and 6MWD have a significance of 0.000. Strictly speaking, this significance does not exist, rather the statistical software shows only the first digits. In this case, it is better to write p<0.001
  3. Patients walk a very low distance. Looking at the data and the standard deviation, the question arises: did all the patients perform the test? or some patients were not able to walk?
  4. Data is usually presented with two decimals. However, it does not make much clinical sense to display these data with so many decimal places. I recommend reducing to 1 or no decimal
  5. The authors found that subjects with polyneuropathy walked less than those without. However, in the evaluation of the groups, patients with polyneuropathies are older, have a lower DLCO and a lower PaO2. We know that the distance walked depends on these 3 variables, it has been shown in different diseases, particularly in pulmonary hypertension. For this reason, it seems very risky to suggest in the discussion (and in the abstract) that neuropathy seems to be one of the limiting factors in exercise capacity. I suggest rewriting this paragraph and stating the differences that I am raising.

Author Response

  1. In the methods section, add the reference of the STROBE checklist

REPLY: we have added the required reference.

  1. In table 1, the values of PaO2, DLCO and 6MWD have a significance of 0.000. Strictly speaking, this significance does not exist, rather the statistical software shows only the first digits. In this case, it is better to write p<0.001

REPLY: We have removed the p-values from table 1 and we graphically have showed significant pairwais comparisons in Figure 1.

  1. Patients walk a very low distance. Looking at the data and the standard deviation, the question arises: did all the patients perform the test? or some patients were not able to walk?

REPLY: 7 patients without PNS and 7 patients with mononeuropathy were not able to perform the basal evaluation.

  1. Data is usually presented with two decimals. However, it does not make much clinical sense to display these data with so many decimal places. I recommend reducing to 1 or no decimal

REPLY: We have changed the values in the Table 1 reporting medians and ranges and we have presented just one decimal as required.

  1. The authors found that subjects with polyneuropathy walked less than those without. However, in the evaluation of the groups, patients with polyneuropathies are older, have a lower DLCO and a lower PaO2. We know that the distance walked depends on these 3 variables, it has been shown in different diseases, particularly in pulmonary hypertension. For this reason, it seems very risky to suggest in the discussion (and in the abstract) that neuropathy seems to be one of the limiting factors in exercise capacity. I suggest rewriting this paragraph and stating the differences that I am raising.

REPLY: We agree with Reviewer. Therefore, we have modified the abstract and re-written the sentence in the discussion. Please read now: “In the present study, we showed that exercise capacity was somehow related to the presence of PNS abnormalities, similarly to what has been observed in COPD patients, since participants without PNS involvement reported significantly higher values of 6MWD when compared to both polyneuropathy and mononeuropathy group. However, considering that the normal patient group was also characterized by a lower age and higher DLCO and PaO2 values, we cannot exclude that the better exercise capacity may depend on these three variables rather than the presence of PNS involvement.”.

Round 2

Reviewer 2 Report

The authors have satisfactorily responded to my comments.